# Lipid-siRNA Conjugates Targeting High PD-L1 Expression as Potential Novel Immune Checkpoint Inhibitors

**DOI:** 10.3390/biom15020293

**Published:** 2025-02-15

**Authors:** Rina Tansou, Takanori Kubo, Haruka Nishida, Yoshio Nishimura, Keichiro Mihara, Kazuyoshi Yanagihara, Toshio Seyama

**Affiliations:** 1Laboratory of Molecular Cell Biology, Department of Life Science, Faculty of Pharmacy, Yasuda Women’s University, Hiroshima 731-0153, Japan; 23331401@st.yasuda-u.ac.jp (R.T.); 20141228@st.yasuda-u.ac.jp (H.N.); kayanag2@ncc.go.jp (K.Y.); seyama@yasuda-u.ac.jp (T.S.); 2School of Pharmaceutical Sciences, Ohu University, Fukushima 963-8611, Japan; y-nishimura@pha.ohu-u.ac.jp; 3Department of International Center for Cell and Gene Therapy, Fujita Health University, Toyoake 470-1192, Japan; keichiro.mihara@fujita-hu.ac.jp; 4Division of Rare Cancer Research, National Cancer Center Research Institute, Tokyo 104-0045, Japan

**Keywords:** siRNA conjugates, programmed death 1 ligand (PD-L1), immune checkpoint inhibitors, nucleic acid drugs, interferon gamma (IFNγ) stimulation

## Abstract

Programmed death 1 ligand (PD-L1), an important immune checkpoint molecule, is mainly expressed on cancer cells and has been shown to exert an immunosuppressive effect on T-cell function by binding to programmed cell death 1 (PD-1) expressed on T-cells. Recently, immune checkpoint inhibitors using antibody drugs such as nivolumab and atezolizumab have attracted attention. However, clinical challenges, including limitations to the scope of their application, are yet to be addressed. In this study, we developed a novel immune checkpoint inhibitor that targets PD-L1 using lipid-siRNA conjugates (lipid-siPDL1s). The inhibitory effect of lipid-siPDL1s on PD-L1 expression was evaluated and found to strongly suppress mRNA expression. Notably, lipid-siPDL1s exerted a significantly stronger effect than unmodified siPDL1. Interestingly, lipid-siPDL1s strongly inhibited PD-L1 expression despite cancer cell stimulation by interferon-gamma, which induced the overexpression of PD-L1 genes. These results strongly suggest that lipid-siPDL1s could be used as novel immune checkpoint inhibitors.

## 1. Introduction

Cancer immunotherapy is a promising treatment option that enhances immune response and induces cancer cell death mediated via the host immune system [1,2]. However, cancer cells resist immune cells via immune checkpoint pathways, inactivating the immune system [3]. This immune checkpoint is highly expressed in numerous cancer cells, with the identification of the B7 group (e.g., B7-H1, B7-H2, B7-H3, and B7-H4) [4]. B7-H1, also known as programmed death 1 ligand (PD-L1), is a transmembrane protein that inhibits immune responses by interacting with programmed death-1 (PD-1), an inhibitory receptor expressed on the T cell surface subsequent to T cell activation [5]. The binding of PD-L1 to PD-1 results in the inhibition of T cell proliferation, cytokine production, and cytolytic activity, thereby inducing the functional inactivation of T cells. Furthermore, cancer cells stimulated by interferon-gamma (IFNγ), a cytokine produced by activated T cells, were shown to exhibit enhanced PD-L1 expression [6,7]. Recently, immune checkpoint inhibitors targeting PD-L1 or PD-1 have garnered considerable attention owing to their remarkable antitumor efficacy [8,9,10,11]. Immune checkpoint inhibitors represent a novel class of immunotherapies that reactivate tumor-specific T-cell immunity, which is suppressed by PD-L1 expression in tumor cells by inhibiting PD-L1 binding. Specifically, anti-PD-L1 antibodies, such as atezolizumab [12], avelumab [13,14], and durvalumab [15], and anti-PD-1 antibodies, such as nivolumab [16,17] and pembrolizumab [18], are antibody drugs approved as immune checkpoint inhibitors that have demonstrated superior therapeutic outcomes to small molecule-based conventional anticancer therapies. Thus, several immune checkpoint inhibitors capable of inducing T-cell reactivation have been developed as antibody drugs and have been applied clinically. Immunotherapy using immune checkpoint inhibitors is currently used to treat various malignancies; however, the overall response rate remains suboptimal [19,20,21]. Therefore, predicting therapeutic efficacy by examining tumor PD-L1 expression and other relevant factors is necessary to ensure that this treatment is administered to patients who are likely to respond well. Additionally, issues such as serious side effects observed in some patients and the high cost of medical care are yet to be addressed. Furthermore, some cancer cells cannot effectively induce T-cell activation despite the use of immune checkpoint inhibitors; this could be because cancer cells use different immune checkpoints.

In this study, we examined the potential of nucleic acid drugs as a novel class of immune checkpoint inhibitors that could surpass the efficacy of conventional antibody-based therapeutics. Nucleic acid drugs are chemically synthesized from DNA and RNA components that target and inhibit gene expression in cells, thereby preventing the translation of pathogenic proteins [22,23]. Nucleic acid therapeutics include antisense oligonucleotides, aptamers, small interfering RNA (siRNAs), and other approved methods for treating intractable diseases [24]. In particular, siRNA-based drugs have attracted considerable attention in recent years [25,26,27,28]. These drugs effectively inhibit target gene expression at low concentrations by forming a protein complex known as the RNA-induced silencing complex within the cell, which then cleaves the target mRNA, leading to potent gene suppression [29,30]. Currently, approved siRNA drugs include Onpattro^®^ (patisiran) for hereditary amyloid transthyretin amyloidosis [31], Givlaari^®^ (givosiran) for acute hepatic porphyria [32], Oxlumo^®^ (lumasiran) for primary hyperoxaluria type I [33], Leqvio^®^ (inclisiran) for hypercholesterolemia and mixed dyslipidemia [34], Amvuttra^®^ (vutrisiran) for hereditary transthyretin-mediated amyloidosis [35], and Rivfloza^®^ (nedosiran) for primary hyperoxaluria [36]. These drugs primarily target rare and difficult-to-treat diseases. Except for Onpattro^®^, all approved siRNA drugs are *N*-acetylgalactosamine (GalNAc)-siRNA conjugates, in which GalNAc is covalently bound to the terminus of the siRNA molecule. GalNAc is a ligand of the asialoglycoprotein receptor (ASGPR). GalNAc-siRNA accumulates in hepatocytes with high levels of ASGPR and is taken up by hepatocytes via this receptor [37]. Other GalNAc-siRNA conjugates are under investigation in clinical trials for several diseases, and further progress is expected. Cholesterol-siRNA (Chol-siRNA) conjugates are being developed for systemic administration, with numerous studies demonstrating their potential applications in the field [38,39]. Chol-siRNA conjugates reportedly show an affinity for low-density lipoprotein, along with an improved tolerance to degradation in the blood [40,41]. Additionally, Chol-siRNA conjugates tend to accumulate in the liver, abdominal cavity, lungs, and kidneys, and intravenous administration of these conjugates against apolipoprotein B suppresses target gene expression in the liver [42]. siRNA conjugates, which combine a ligand with siRNA, can confer properties of the ligand to the siRNA, potentially addressing challenges associated with siRNA drugs. These challenges include enhanced blood retention, efficient cellular uptake, and precise delivery to the target tissues. We have focused on fatty acids as ligands and successfully conjugated several saturated and unsaturated fatty acids to siRNAs using our innovative conjugate synthesis method [43]. Previous studies have demonstrated that lipid-siRNA conjugates (lipid-siRNAs), in which a fatty acid is covalently linked to the 5′-terminus of the sense strand of siRNA, exhibit potent RNA interference (RNAi) effects both in vitro and in vivo [44,45,46,47]. Recently, we reported that systemically administered lipid-siRNAs targeting β-catenin could robustly suppress the growth of tumors that metastasized to the liver in a mouse model of pancreatic cancer [48].

Various cancer cell types exhibit differential expression of cancer immune checkpoints, including PD-L1, and IFNγ stimulation substantially modulates PD-L1 expression. In this study, we aimed to suppress the expression of immune checkpoint molecules, especially PD-L1, on the surface of cancer cells using RNAi technology with lipid-siRNA. Our approach was designed to be effective regardless of mRNA and protein levels, cancer cell type, or changes in expression levels upon IFNγ stimulation. This study could elucidate the potential of lipid-siRNAs as a novel immune checkpoint inhibitor based on siRNA therapeutics.

## 2. Materials and Methods

### 2.1. Cell Culture

T47D, a breast cancer cell line, and A549, a lung cancer cell line, were obtained from the Riken Bioresource Center Cell Bank (Tsukuba, Japan). The 44As3 cell line, originating from HSC-44PE cells in the pleural fluid of a patient with scirrhous gastric cancer, was established by selecting clones capable of inducing ascites in mice [49,50]. A549 and 44As3 cell lines were cultured in RPMI-1640 medium (Wako, Osaka, Japan), and T47D cells were cultured in DMEM (Wako) supplemented with 10% fetal bovine serum (FBS; GIBCO BRL, Grand Island, NY, USA), 100 IU/mL penicillin G sodium, and 100 μg/mL streptomycin sulfate (Wako). Cultures were maintained in a humidified incubator at 37 °C and 5% CO_2_. mRNA expression levels of PD-L1 and B7-H4 were analyzed in A549, T47D, and 44As3 cell lines by reverse transcription-quantitative polymerase chain reaction (RT-qPCR). Western blotting was performed to determine PD-L1 expression at the protein level. The cell lines were stimulated with IFNγ (Thermo Fisher Scientific, Waltham, MA, USA) to enhance PD-L1 expression. To investigate the effect of IFNγ, the final concentrations of IFNγ were adjusted to 0, 6.25, 12.5, 25, 50, and 100 ng/mL, and PD-L1 expression was evaluated after 24 h by RT-qPCR. IFNγ was established at a final concentration of 100 ng/mL to assess the RNAi effects of lipid-siPDL1s.

### 2.2. RT-qPCR

Total RNA was extracted using an RNeasy Mini Kit (Qiagen, Valencia, CA, USA), and the yield was determined based on the optical density measured at 260 nm using NanoDrop One^c^ (Thermo Fisher Scientific). The primer sequences used for target mRNAs were as follows: PD-L1 forward primer, 5′-TGCCGACTACAAGCGAATTACTG-3′; PD-L1 reverse primer, 5′-CTGCTTGTCCAGATGACTTCGG-3′; B7-H4 forward primer, 5′-CACCAGGATAACATCTCTCAGTGAA-3′; B7-H4 reverse primer, 5′-TGGCTTGCAGGGTAGAATGA-3′; GAPDH forward primer, 5′-ACGACCAAATCCGTTGACTC-3′; GDPDH reverse primer, 5′-GCTCTCTGCTCCTCCTGTTC-3′. RT-qPCR was performed using a Thermal Cycler Dice Real-Time System III (Takara Bio, Shiga, Japan) with a Luna Universal one-step qPCR kit (New England BioLabs, Ipswich, MA, USA) according to the manufacturer’s protocol.

### 2.3. Western Blotting Analysis

Briefly, A549, T47D, and 44As3 cells grown to confluence in 6-well plates were lysed using 500 μL NP40 buffer (Wako) containing 1% protease inhibitors (Wako). Lysis was performed on ice for 30 min, and samples were then centrifuged at 1500 rpm for 10 min. Proteins were separated by 7.5% sodium dodecyl sulfate (SDS)-polyacrylamide gel electrophoresis (PAGE; Nacalai Tesque, Kyoto, Japan). The separated proteins were subsequently transferred onto polyvinylidene difluoride membranes (Merck, Darmstadt, Germany). The membranes were blocked with Blocking One (Nacalai Tesque) for 30 min at room temperature and subsequently incubated overnight at 4 °C with specific antibodies (1:1000) against PD-L1 (E1L3N^®^ XP^®^ Rabbit mAb [HRP Conjugate], Cell Signaling Technology, MA, USA), β-actin (Monoclonal Anti-β-Actin antibody, Sigma-Aldrich, St. Louis, MO, USA). Subsequently, the membranes were incubated with horseradish peroxidase-conjugated secondary antibodies (1:10,000), specifically anti-mouse (Cytiva, Wilmington, DE, USA). The resultant signals were visualized using the ECL Prime Western blotting Detection Reagent (Cytiva).

### 2.4. Design and Synthesis of Lipid-siRNAs

siRNAs comprising 21 base pairs with a 2-nucleotide extension at the 3′ end were designed to specifically target PD-L1 (siPDL1) and B7-H4 (siB7H4) genes. A control siRNA (siCtrl; Santa Cruz Biotechnology, Dallas, TX, USA) that did not show homology with any eukaryotic genes was used as a negative control. siRNAs or single-stranded RNAs (ssRNAs) with amino acid modifications were purchased from Dharmacon (Lafayette, CO, USA) or Integrated DNA Technologies (IDT, Coralville, IA, USA). Palmitic acid (C16), stearic acid (C18), oleic acid (Ole), and linoleic acid (Lio) were conjugated to the 5′-end of the sense strand via the amino moieties of the single-stranded RNA. The carbonyl groups of all fatty acids were esterified with *N*-hydroxysuccinimide (NHS) ester, which exhibited high reactivity. Briefly, amino-modified ssRNAs (1 mM in water) were reacted with each fatty acid-NHS ester (100 mM in *N*,*N*-dimethylformamide) in isopropanol (Wako) containing 0.5–1 μL *N*,*N*-diisopropylethylamine (Wako) for 16–24 h at room temperature. The molar ratio of amino-modified ssRNAs to fatty acids was adjusted to 1:100 or 1:50. Lipid-conjugated ssRNAs were purified using reverse-phase high-performance liquid chromatography (RP-HPLC; Shimadzu, Kyoto, Japan). This procedure utilized an octadecylsilyl column (4.6 × 75 mm, 5 μm; Osaka Soda, Osaka, Japan) and a linear gradient ranging from 7 to 70% acetonitrile (Wako) in 20 mM triethylammonium acetate (Wako) over a 40-min duration (pH 7.0). The purity and molecular weight of the conjugates were confirmed using RP-HPLC and matrix-assisted laser desorption/ionization-time of flight mass spectrometry (MALDI-TOF MS) (MALDI-7090; Shimadzu, Kyoto, Japan). The conjugate yields were determined by spectrophotometric analysis based on absorption at 260 nm using a V-670 spectrophotometer (JASCO, Tokyo, Japan). Double-stranded lipid-conjugated siRNAs were synthesized by combining lipid-conjugated ssRNAs with equimolar quantities of single-antisense stranded RNA (asRNA). Subsequently, the resultant unmodified and lipid-conjugated siRNAs were subjected to quality assessment and dimensional analysis using 20% PAGE. siRNAs conjugated to siPDL1 and siB7H4 with C16, C18, Ole, and Lio were designated as C16-siPDL1, C16-siB7H4, C18-siPDL1, C18-siB7H4, Ole-siPDL1, Ole-siB7H4, Lio-siPDL1, and Lio-siB7H4, respectively.

### 2.5. Lipid-siPDL1s-Induced Suppression of Immune Checkpoint Gene Expression

To assess the RNAi effect of unmodified siRNAs and lipid-siRNAs targeting PD-L1 (siPDL1 and lipid-siPDL1s) and B7-H4 (siB7H4 and lipid-siB7H4s), experiments were performed using Lipofectamine RNAiMAX (LFRNAi; Thermo Fisher Scientific). Briefly, A549, T47D, and 44As3 cells were cultured at a concentration of 5 × 10^4^ cells per 0.5 mL RPMI-1640 medium in each well of a 48-well plate. The cells were incubated at 37 °C in an atmosphere containing 5% CO_2_ and 95% air. siRNAs and lipid-siRNAs were pre-incubated with LFRNAi (0.75 μL/well) in 25 μL/well Opti-MEM (Thermo Fisher Scientific) for 20 min. Next, 25 μL of the mixture was added to each well of a 48-well microplate, followed by the addition of 225 μL of OptiMEM per well; the final concentration of siRNA or lipid-siRNAs was 50 nM. siCtrl was used as a negative control and subjected to identical treatments. The culture medium was replaced with fresh culture medium 24 h after the addition of siRNAs and lipid-siRNAs, and cells were cultured in a humidified atmosphere (5% CO_2_, 37 °C) for 24, 48, and 72 h for RNAi analysis. The in vitro RNAi potency of siRNA and lipid-siRNAs against PD-L1 and B7-H4 was evaluated by quantitatively measuring their mRNA levels using RT-qPCR. The RNAi potency of siRNA and lipid-siRNAs was assessed as a percentage of control cells treated with siCtrl after each sample was calibrated using glyceraldehyde-3-phosphate dehydrogenase (GAPDH) as an internal control. To examine RNAi effects in the PD-L1 high expression state induced by IFNγ stimulation, cells were treated with siPDL1 and lipid-siPDL1s following the aforementioned protocol, with the IFNγ concentration consistently maintained at 100 ng/mL.

### 2.6. Lipid-siPDL1s-Induced Inhibition of Immune Checkpoint Protein Expression

To visualize PD-L1 expression on the cell surface, A549, T47D, and 44As3 cells were stained with Alexa-488 labeled anti-PD-L1 antibody (Alexa Fluor^®^ 488 Anti-PD-L1 antibody; Abcam, Cambridge, UK). Stained cells were observed under a confocal fluorescence microscope (FV1000; Evident, Tokyo, Japan). A549, T47D, and 44As3 cells were pre-seeded on 35 mm glass-bottom dishes (AGC Techno Glass, Shizuoka, Japan) and incubated overnight at 37 °C. Each cancer cell line was transfected with siPDL1 and lipid-siPDL1s using LFRNAi. Subsequently, Alexa Fluor^®^ 488 Anti-PD-L1 antibody was used to assess the cell surface expression of PD-L1 at 48 and 96 h post-transfection. The concentration of siPDL1s and lipid-siPDL1s was 50 nM, and transfection was performed as described in the experiment for siRNA effects described above. siCtrl was used as a negative control and subjected to identical treatments. Cells were prepared for confocal fluorescence microscopy as follows: first, the cells were washed three times with PBS (-) and fixed with 4% paraformaldehyde phosphate buffer solution (Wako) for 15 min. The cells were then washed three times with 0.5% bovine serum albumin (BSA; Wako)/PBS (-) solution and incubated with Alexa Fluor^®^ 488 Anti-PD-L1 antibody diluted 1:100 in 5% BSA/PBS (-) solution at room temperature for 1 h. Simultaneously, cells were stained with MitoTracker Red CMXRos (Thermo Fisher Scientific) and Hoechst 33342 (Thermo Fisher Scientific) to visualize the mitochondria and nucleus. The cells were subsequently washed three times with PBS (-), and their respective fluorescence signals were observed using a confocal fluorescence microscope. Hoechst, Alexa-488, and MitoTracker fluorescent signals were excited using 405, 473, and 559 nm lasers, respectively.

Western blotting was performed to analyze the siPDL1- and lipid-siPDL1s-mediated suppression of immune checkpoint protein expression in cells. The concentration of siPDL1 and lipid-siPDL1s was 50 nM, and transfection was performed as described above. The same treatment was performed using siCtrl as the negative control. Cells were subsequently lysed at 48 and 96 h post-transfection, and PD-L1 protein expression was verified in each sample by western blotting analysis described above.

### 2.7. Statistical Analysis

Data were analyzed using unpaired *t*-tests. Data values are expressed as mean ± standard deviation (SD). A *p* value < 0.05 was deemed statistically significant.

## 3. Results

### 3.1. Synthesis of Lipid-siRNAs

We designed siRNAs (siPDL1 and siB7H4) targeting PD-L1 and B7-H4, respectively. Lipid-ssRNAs (lipid-ssPDL1s and lipid-ssB7H4s) were synthesized by covalently attaching fatty acids to the 5′-terminus of the sense strand of designed siRNAs (Figure 1A and Appendix A). The fatty acids conjugated to ssRNAs included palmitic, stearic, oleic, and linoleic acids. The synthesized lipid-ssRNAs were purified using RP-HPLC, resulting in high-purity lipid ssRNAs (Figure 1B and Appendix A). High-performance liquid chromatography analysis using an octadecylsilyl column revealed that elution times varied according to the properties of the attached fatty acids. C18-ssRNA bound to stearic acid, a saturated fatty acid with an extended carbon chain, exhibited the longest elution time (32.4 min), whereas Lio-ssRNA bound to linoleic acid, an unsaturated fatty acid containing 18 carbon atoms, had the shortest elution time (28.7 min). The molecular weights of purified lipid-ssRNAs were analyzed using MALDI-TOF MS and were in accordance with the calculated values (Table 1 and Appendix A). Synthesized lipid-ssRNAs were annealed using antisense strands to produce double-stranded lipid-siRNAs (lipid-siPDL1s and lipid-siB7H4s). The sequences of siPDL1 and siB7H4, as well as the structures of the lipid-siRNAs, are illustrated in Figure 1C and Appendix A. PAGE was performed to analyze the purity of each lipid-siRNAs (Figure 1D). In PAGE analysis, lipid-siRNAs showed an upward movement based on their molecular weights. We evaluated the effects of these lipid-siRNAs on PD-L1 and B7-H4 expression in A549, T47D, and 44As3 cancer cell lines.

### 3.2. Differential Expression of Immune Checkpoints in Cancer Cells

We investigated differential mRNA expression of PD-L1 and B7-H4 in three cancer cell lines: lung cancer (A549), breast cancer (T47D), and scirrhous gastric cancer (44As3). RT-qPCR analysis was performed to quantify the relative expression levels of PD-L1 and B7-H4 mRNA in T47D and 44As3 cells compared with those in A549 cells (Figure 2 and Appendix A). Characteristic immune checkpoint expression was observed in all three cancer cell lines. Regarding PD-L1 mRNA expression, 44As3 cells exhibited the highest levels among the three cancer cell lines, with approximately 20-fold higher expression than A549 cells (Figure 2). On the other hand, PD-L1 expression in T47D cells was almost similar to that observed in A549 cells. Upon comparing the expression of B7-H4 mRNA in the three cancer cell lines, T47D cells exhibited approximately 10-fold higher B7-H4 mRNA expression than A549 cells (Appendix A). Conversely, 44As3 cells, with a notably high PD-L1 expression, had significantly reduced B7-H4 mRNA levels—approximately 0.35 times lower than in A549 cells. Herein, we synthesized siRNAs and lipid-siRNAs targeting PD-L1 and B7-H4 and examined their RNAi effects in three cell lines.

### 3.3. RNAi Efficacy of Lipid-siRNAs Targeting PD-L1 and B7-H4 in Cancer Cells

The effects of siRNAs and lipid-siRNAs on PD-L1 and B7-H4 expression in A549, T47D, and 44As3 cells were investigated (Figure 3 and Appendix A). The knockdown effects were evaluated by performing RT-qPCR 24 h after the transfection of target cells. Based on the results, siPDL1 suppressed PD-L1 expression by approximately 60% in A549 and T47D cells and by approximately 50% in 44As3 cells compared with the control (siCtrl). Lipid-siPDL1s demonstrated superior gene knockdown effects than siPDL1 in the A549, T47D, and 44As3 cell lines. Lipid-siPDL1s exerted the most pronounced suppression of PD-L1 gene expression in T47D cells, reducing expression by 80-90% compared with siCtrl-treated cells. Lipid-siPDL1s also exhibited potent RNAi efficacy against both the A549 and 44As3 cell lines, reducing PD-L1 gene expression by approximately 70%. In addition, we examined the RNAi effects on B7-H4 expression. siB7H4 and lipid-siB7H4s induced markedly potent RNAi effects in downregulating B7-H4 gene expression in the A549, T47D, and 44As3 cell lines (Appendix A). siB7H4 suppressed B7-H4 mRNA expression by approximately 70% in A549, T47D, and 44As3 cells, whereas lipid-siB7H4s suppressed B7-H4 mRNA expression by approximately 80%, exerting a more potent RNAi effect than siB7H4. Thus, lipid-siRNAs exhibited enhanced RNAi effects compared with unmodified siRNA across all cell types and target genes examined in this study.

### 3.4. IFNγ Stimulation Induces Increased PD-L1 Expression in Cancer Cells

We investigated the variation in PD-L1 expression in A549, T47D, and 44As3 cell lines upon IFNγ stimulation (Figure 4). IFNγ stimulation upregulated PD-L1 expression in all three cancer cell lines. Stimulation with 100 ng/mL IFNγ resulted in approximately a 5- to 20-fold increase in PD-L1 expression than that under unstimulated conditions. Even exposure to low concentrations of IFNγ (6.25 ng/mL) also increased PD-L1 expression in cancer cells. Notably, A549 and T47D cells, which originally expressed relatively low PD-L1 levels, showed up to a 20-fold increase in PD-L1 expression following IFNγ stimulation at 100 ng/mL concentration. Furthermore, 44As3 cells, which originally expressed high PD-L1 levels, showed up to a 5-fold increase in PD-L1 expression under the same conditions (Figure 4A). This difference in sensitivity to IFNγ between cancer cells is dependent on the baseline PD-L1 expression level. Specifically, PD-L1 expression in 44As3 cells, which initially exhibit high PD-L1 levels, does not fluctuate greatly upon IFNγ stimulation, whereas A549 and T47D cells, which initially have low PD-L1 levels, show a significant increase in PD-L1 expression in response to IFNγ stimulation. Interestingly, 44As3 exhibited the highest PD-L1 expression among the three cancer cell lines following IFNγ stimulation at 100 ng/mL with levels approximately 15-fold higher than those observed in A549 and T47D cells (Figure 4B). Additionally, T47D cells exhibited PD-L1 expression patterns comparable to those of A549 cells upon IFNγ stimulation. We performed western blot analysis to determine PD-L1 protein levels. Consistent with the RT-qPCR results, PD-L1 protein levels were elevated in IFNγ-stimulated cells compared with those in non-stimulated cells, with 44As3 cells exhibiting the highest PD-L1 protein levels upon IFNγ stimulation (Figure 4C). Accordingly, the RNAi effects of lipid-siPDL1s were examined at the highest PD-L1 expression level in each cancer cell line stimulated with 100 ng/mL IFNγ.

### 3.5. Inhibition of PD-L1 mRNA Expression Using Lipid-siPDL1s Under Upregulated PD-L1 Conditions

To investigate the RNAi efficacy of siPDL1 and lipid-siPDL1s under conditions of elevated PD-L1 expression, we stimulated A549, T47D, and 44As3 cell lines with IFNγ at a concentration of 100 ng/mL. RT-qPCR analysis was performed to evaluate the suppression of PD-L1 expression in each cell type at 24, 48, and 72 h post-treatment with siPDL1 and lipid-siPDL1s (Figure 5). In all cell lines, siPDL1 and lipid-siPDL1s significantly inhibited PD-L1 expression compared with siCtrl, and this effect persisted for up to 72 h post-treatment. In A549 cells, which exhibited PD-L1 expression elevated by ~20-fold upon IFNγ stimulation at 100 ng/mL, siPDL1 inhibited PD-L1 expression by approximately 60–70% at 24 and 48 h post-treatment and by approximately 80% at 72 h post-treatment compared with siCtrl (Figure 5A). Lipid-siPDL1s demonstrated stronger RNAi effects and superior persistence, suppressing PD-L1 expression by approximately 80% at 24 and 48 h post-treatment and by more than 90% at 72 h post-treatment, exceeding the gene expression suppression effect of siPDL1 under all experimental conditions. In T47D cells, a similar 20-fold increase in PD-L1 expression was observed following IFNγ stimulation, as evidenced in A549 cells; siPDL1 treatment suppressed PD-L1 expression by approximately 50, 45, and 40% at 24, 48, and 72 h, respectively, compared with siCtrl, with the effect diminishing gradually (Figure 5B). Conversely, lipid-siPDL1s significantly suppressed PD-L1 expression in T47D cells, reducing expression by 70–80% within 24–72 h post-treatment, with no decline in RNAi efficacy over extended incubation. In 44As3 cells, which typically exhibit high PD-L1 levels and a 5-fold increase upon IFNγ stimulation, both siPDL1 and lipid-siPDL1s significantly inhibited PD-L1 expression (Figure 5C). Application of siPDL1 to 44As3 cells reduced PD-L1 expression by approximately 60% at 24 h and 70% at both 48 and 72 h post-treatment compared with siCtrl. Lipid-siPDL1s exerted a stronger RNAi effect than siPDL1 on 44As3 cells, suppressing PD-L1 expression by approximately 80% at 24–72 h after treatment.

Thus, despite differences in the effects depending on the nature of cells with distinct IFNγ sensitivities, both siPDL1 and lipid-siPDL1s strongly suppressed targeted PD-L1 expression. In particular, lipid-siPDL1s exerted a more significant RNAi effect than siPDL1 under all conditions.

### 3.6. Suppression of Elevated Cell Surface PD-L1 Protein Expression by Lipid-siPDL1s

The inhibitory effect of siPDL1 and lipid-siPDL1s on PD-L1 protein expression in cancer cells upon IFNγ stimulation was examined by performing confocal fluorescence microscopy (Figure 6) and western blot analysis (Figure 7). In both analyses, siPDL1 and lipid-siPDL1s were transfected with LFRNAi and subsequently evaluated 48 and 96 h post-transfection.

Confocal microscopy analysis was performed to assess PD-L1 protein expression in A549, T47D, and 44As3 cell lines following IFNγ stimulation using fluorescent-labeled anti-PD-L1 antibodies for visualization (Figure 6). According to confocal microscopy analysis, the relative expression level of PD-L1 mRNA in each cancer cell line correlated with the fluorescence intensity of the anti-PD-L1 antibody, with a lower anti-PD-L1 antibody fluorescence intensity detected in siCtrl-treated A549 cells and a higher intensity detected in 44As3 cells. In A549 cells, siPDL1 and lipid-siPDL1s significantly reduced PD-L1 expression at 48 and 96 h post-transfection. Cells treated with lipid-siPDL1s conjugated with stearic, oleic, and linoleic acids (18 carbon chains) exhibited almost no anti-PD-L1 antibody-induced fluorescence at 48 and 96 h, indicating a significant reduction in PD-L1 protein on A549 cell surfaces. siCtrl-treated T47D cells exhibited marked anti-PD-L1 antibody fluorescence intensity, suggesting that PD-L1 was highly expressed on the surface of T47D cells. Compared with siCtrl-treated cells, T47D cells transfected with siPDL1 and lipid-siPDL1s exhibited reduced anti-PD-L1 antibody fluorescence intensity. Lipid-siPDL1s-treated cells exhibited significantly reduced anti-PD-L1 antibody fluorescence intensity compared with siPDL1-treated cells, indicating that lipid-siPDL1s strongly suppressed PD-L1 protein on the surface of T47D cells. The knockdown effect differed with incubation time, with the anti-PD-L1 antibody fluorescence decreasing at 48 h post-transfection compared with that at 96 h post-transfection. In 44As3 cells, which exhibited higher PD-L1 mRNA expression than other cells, significant anti-PD-L1 antibody fluorescence intensity was observed at the cell membrane of siCtrl-treated cells. Both siPDL1 and lipid-siPDL1s induced notable RNAi effects in 44As3 cells, leading to reduced fluorescence intensity near the cell membrane of treated cells. In particular, lipid-siPDL1s-treated cells exhibited a significant reduction in fluorescence around the cell membrane, indicating reduced PD-L1 expression on the 44As3 cell surface. Moreover, the anti-PD-L1 antibody fluorescence intensity diminished 96 h after lipid-siPDL1s treatment, revealing that lipid-siPDL1s prolonged the PD-L1 knockdown effect.

We performed western blotting analysis to evaluate the expression of PD-L1 protein in each sample, normalizing expression to that of β-actin as an endogenous control protein (Figure 7). siCtrl-transfected cells exhibited strong PD-L1 protein expression, as confirmed by a strong signal in western blot analysis. PD-L1 protein expression was significantly suppressed in siPDL1- and lipid-siPDL1s-transfected cells compared with that in siCtrl-transfected cells, and a weak signal was detected using western blot analysis. In addition, lipid-siPDL1s-transfected cells exhibited relatively weaker PD-L1 protein signals than siPDL1-transfected cells, which was almost consistent with the results of suppressed PD-L1 mRNA expression determined by RT-qPCR analysis and PD-L1 protein expression by confocal microscopy. Western blotting further corroborated the sustained lipid-siPDL1s-induced inhibition of PD-L1 protein expression, indicating the efficacy of lipid-siPDL1s. Accordingly, confocal microscopy and western blotting analyses demonstrated that siPDL1 and lipid-siPDL1s reduced PD-L1 mRNA and protein expression in cancer cells.

## 4. Discussion

In this study, we analyzed changes in the expression of immune checkpoints (PD-L1 and B7H4) in lung (A549), breast (T47D), and scirrhous gastric (44As3) cancer cells, along with the inhibitory effects of siRNAs and lipid-siRNAs on these checkpoints. Furthermore, we verified the RNAi effect of lipid-siPDL1s under IFNγ-induced elevated PD-L1 expression and examined its potential as a novel immune checkpoint inhibitor.

Herein, we examined the expression of immune checkpoints in A549, T47D, and 44As3 cell lines and found that cancer cells can modulate their utilization of immune checkpoints in a complex manner. For example, T47D cells have relatively low expression of PD-L1 but high expression of B7-H4, while 44As3 cells exhibit high PD-L1 expression but low B7-H4 expression. In our previous studies, we demonstrated that 44As3 cells represent a relatively high-grade scirrhous gastric carcinoma and induce peritoneal dissemination when transplanted orthotopically into nude mice [49,50]. These highly malignant 44As3 cells exhibited a notable upregulation in PD-L1 expression compared with other cancer cells. These findings have important implications for the selection of immune checkpoint inhibitors. In this study, we also investigated IFNγ-stimulated changes in PD-L1 expression in each cancer cell line. Despite variations in IFNγ sensitivity among the three cancer cell lines, PD-L1 expression was upregulated in all examined cells. The expression of PD-L1 in each cancer cell type was upregulated even at low concentrations of IFNγ (6.25 ng/mL), suggesting that cancer cells are sensitive to IFNγ stimulation. These findings indicate that despite exhibiting low PD-L1 expression, cancer cells can manifest substantial PD-L1 levels in vivo upon stimulation with T cell-derived IFNγ, consequently circumventing T cell-mediated attacks, which has important implications for selecting immune checkpoint inhibitors. In vivo IFNγ stimulation can substantially elevate tumor PD-L1 expression, confounding the assessment of therapeutic efficacy and selection of antibody drugs based on PD-L1 and other tumor factors used as immune checkpoint inhibitors in clinical settings.

Therefore, we focused on siRNA-mediated target mRNA cleavage and hypothesized that robust knockdown of PD-L1 mRNA, with or without IFNγ stimulation, would also suppress PD-L1 expression on cancer cell surfaces. We initially evaluated the efficacy of PD-L1 mRNA knockdown by siPDL1 and lipid-siPDL1s in cancer cells without IFNγ stimulation. The results demonstrated that PD-L1 mRNA expression was downregulated in all cancer cell lines, with lipid-siPDL1s eliciting a more pronounced effect than siPDL1. In addition, siB7H4 and lipid-siB7H4s potently knocked down B7-H4 expression, with lipid-siB7H4s showing superior efficacy. siRNAs targeting mRNAs can be easily designed to match the target and, as demonstrated in this study, exert a strong knockdown effect regardless of whether the target is PD-L1 or B7-H4. Our previous studies have revealed that lipid-siRNAs exhibit a stronger knockdown effect than unmodified siRNAs [44]. In particular, among the 16 types of lipid-siRNAs developed to date, C16-siRNA, C18-siRNA, Ole-siRNA, and Lio-siRNA exhibit potent RNAi effects [43]. Previous studies have shown that lipid-siRNAs conjugated with fatty acids containing 16–18 carbons exhibit moderate interactions with LFRNAiMAX and superior cellular transduction, indicating efficient release into cells, in contrast to lipid-siRNA conjugated with long-chain fatty acids and trans-fatty acids. Therefore, in this study, we investigated the inhibitory effects of C16-siRNA, C18-siRNA, Ole-siRNA, and Lio-siRNA on PD-L1 expression in each cancer cell line. The lipid-siPDL1s and lipid-sB7H4s used in the present study were validated as siRNA conjugates, demonstrating strong RNAi-induced knockdown in cancer cells, surpassing the effects of unmodified siPDL1 and siB7H4. Furthermore, the lipid-siPDL1-mediated knockdown effect remained robust even upon stimulation of cancer cells with IFNγ to induce high PD-L1 expression, consistently reducing PD-L1 mRNA levels regardless of its expression levels. Additionally, the knockdown of PD-L1 mRNA expression in cancer cells exhibited sustained efficacy, with lipid-siPDL1s demonstrating robust RNAi effects even in the presence of elevated PD-L1 mRNA expression following prolonged IFNγ stimulation. Lipid-siPDL1s markedly suppressed PD-L1 protein expression across all three cancer cell types, even when IFNγ stimulation enhanced PD-L1 expression, with the inhibitory effect lasting for at least 96 h.

Recently, Ganesh et al. investigated an in vivo combination therapy using standard-of-care immune checkpoint inhibitors alongside chemically stabilized acylated siRNAs targeting signal transducer and activator of transcription 3 and PD-L1 genes, reporting excellent antitumor effects [51]. This study provides substantial evidence supporting the potential of siRNA conjugates, including lipid-siRNAs, as novel immune checkpoint inhibitors targeting immune checkpoints such as PD-L1. Furthermore, our research group has recently achieved success in synthesizing lipid-siRNA conjugates incorporating chemically modified nucleic acids, such as 2′-O-M and 2′-F. Further development of this research is expected in the future.

## 5. Conclusions

Cancer cells may adeptly employ various immune checkpoint molecules, including PD-L1 and B7-H4. PD-L1 expression is further upregulated by IFNγ stimulation. Thus, the interplay of factors like immune checkpoint types expressed in tumors, their baseline expression levels, and changes with and without IFNγ stimulation can complicate cancer therapy using immune checkpoint inhibitors such as antibody-based drugs. In this study, we investigated the inhibitory effects of lipid-siPDL1s on PD-L1 gene and protein expression in various cancer cells. Lipid-siPDL1 strongly suppressed PD-L1 expression at both mRNA and protein levels, regardless of cancer cell type or baseline PD-L1 expression. This mRNA targeting approach using lipid-siRNA conjugates effectively could suppress all B7 family members, including PD-L1, for a prolonged period, regardless of cancer type or immune checkpoint expression levels. Thus, lipid-siRNA-based technologies hold the potential to be developed as novel immune checkpoint inhibitors capable of replacing traditional antibody-based inhibitors.

## Figures and Tables

**Figure 1 biomolecules-15-00293-f001:**
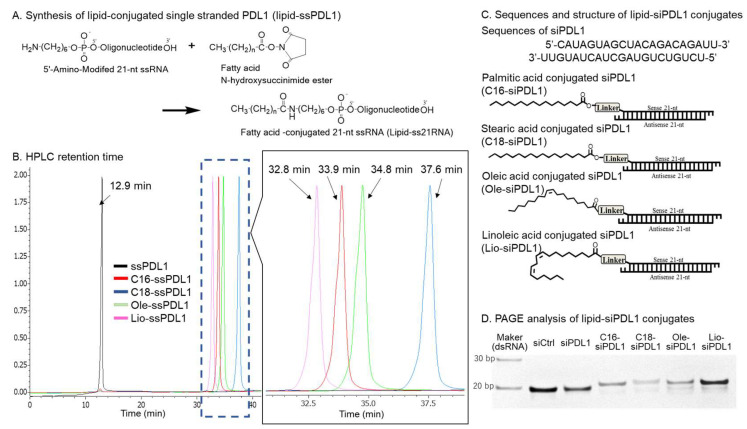
Synthesis (**A**), reverse-phase high-performance liquid chromatography (RP-HPLC) profiles (**B**), sequences and structures (**C**), and polyacrylamide gel electrophoresis (PAGE) analysis of lipid-siPDL1s targeting PD-L1 mRNA (**D**). Lipid-ssPDL1s were synthesized using a simple conjugation method [43]. Fatty acids were conjugated to the 5′-end of the sense strand via an amino linker. RP-HPLC was performed using an octadecylsilyl column (4.6 × 75 mm I.D., 5 μm) and a linear gradient of acetonitrile at concentrations varying from 7 to 70% over 40 min in 100 mM triethylamine acetate (pH 7.0). The elution time depends on the nature of conjugated lipids. The purified lipid-ssPDL1 is double-stranded with the antisense strand, and PAGE analysis confirmed sufficient purity of the resulting lipid-siPDL1s. Original western blot images can be found at Appendix A.

**Figure 2 biomolecules-15-00293-f002:**
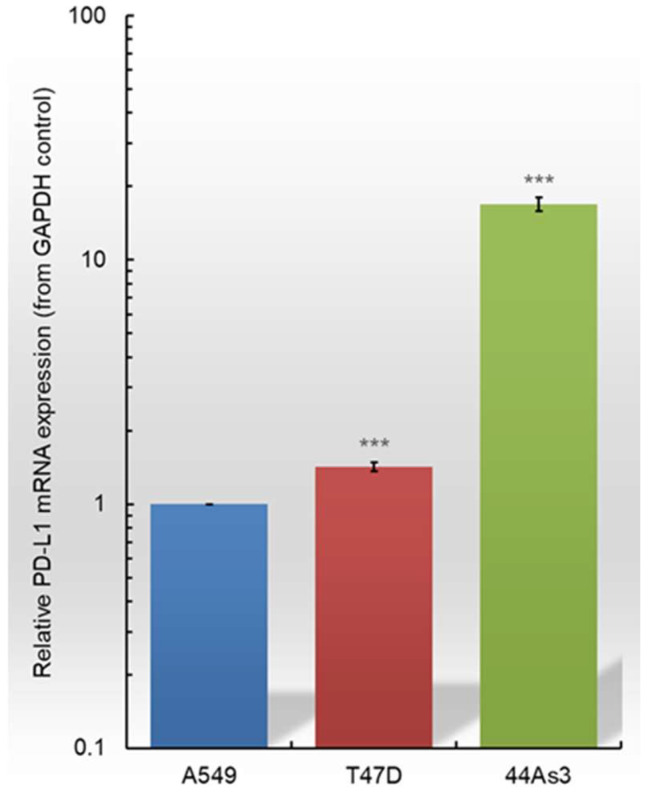
Relative PD-L1 mRNA expression in A549, T47D, and 44As3 cells. PD-L1 mRNA was evaluated in A549 cells and compared with that in T47D and 44As3 cell lines using RT-qPCR. Cancer cell lines exhibit differential PD-L1 expression. Comparing the three cancer cell lines, PD-L1 expression is the highest in 44As3 cells. Data are presented as the mean ± SD of three independent experiments (*** *p* < 0.001 vs. A549; *t*-test). PD-L1, Programmed death 1 ligand; RT-qPCR, reverse transcription-quantitative PCR.

**Figure 3 biomolecules-15-00293-f003:**
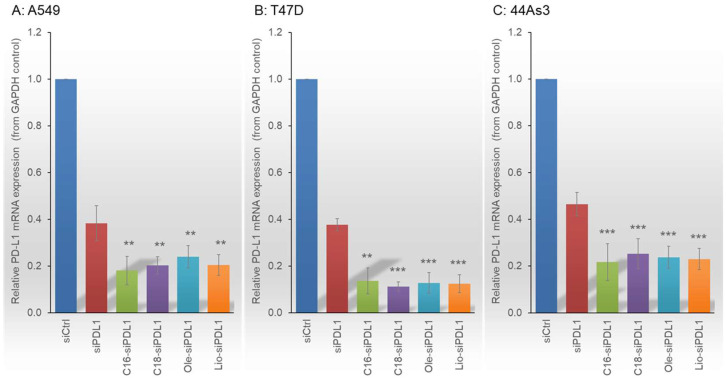
RNAi efficacy of siPDL1 and lipid-siPDL1s against A549 (**A**), T47D (**B**), and 44As3 (**C**) cells. The RNAi effects of siPDL1 and lipid-siPDL1s vary across the different cell types. siPDL1 and lipid-siPDL1s suppress PD-L1 expression by 60–80% in A549, 60–90% in T47D, and 50–80% in 44As3 cells compared to siCtrl-treated cells. For all cell lines, lipid-siPDL1s exhibit stronger RNAi effects than siPDL1. Data are presented as the mean ± SD of 3–5 independent experiments (** *p* < 0.01, *** *p* < 0.001 vs. siPDL1; *t*-test). PD-L1, Programmed death 1 ligand; RNAi, RNA interference.

**Figure 4 biomolecules-15-00293-f004:**
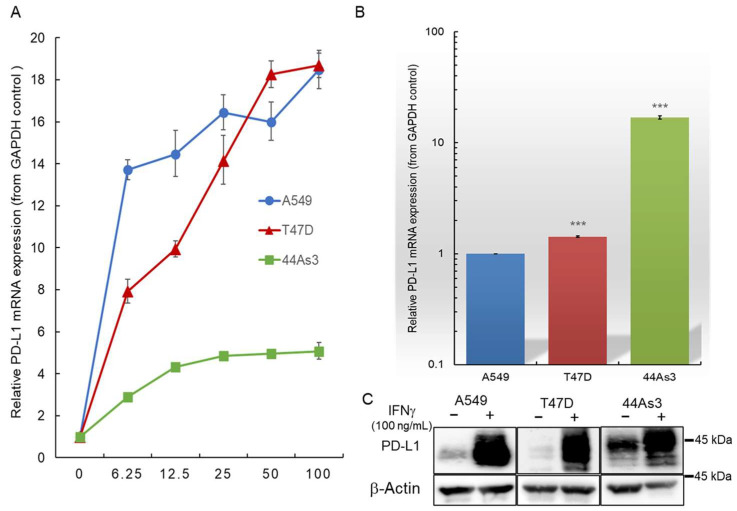
Alterations in PD-L1 expression in A549, T47D, and 44As3 cell lines following IFNγ stimulation. A549, T47D, and 44As3 cells exhibit a 5- to 20-fold increase in PD-L1 expression upon IFNγ stimulation (**A**). Among the three cell types, 44As3 cells exhibit the highest PD-L1 mRNA expression (**B**), even upon IFNγ stimulation. Western blot analysis (**C**) demonstrates increased PD-L1 protein expression in each cancer cell line following IFNγ stimulation. Data are presented as the mean ± SD of three independent experiments (*** *p* < 0.001 vs. A549; *t*-test). IFNγ, interferon-gamma; PD-L1, Programmed death 1 ligand. Original western blot images can be found at Appendix A.

**Figure 5 biomolecules-15-00293-f005:**
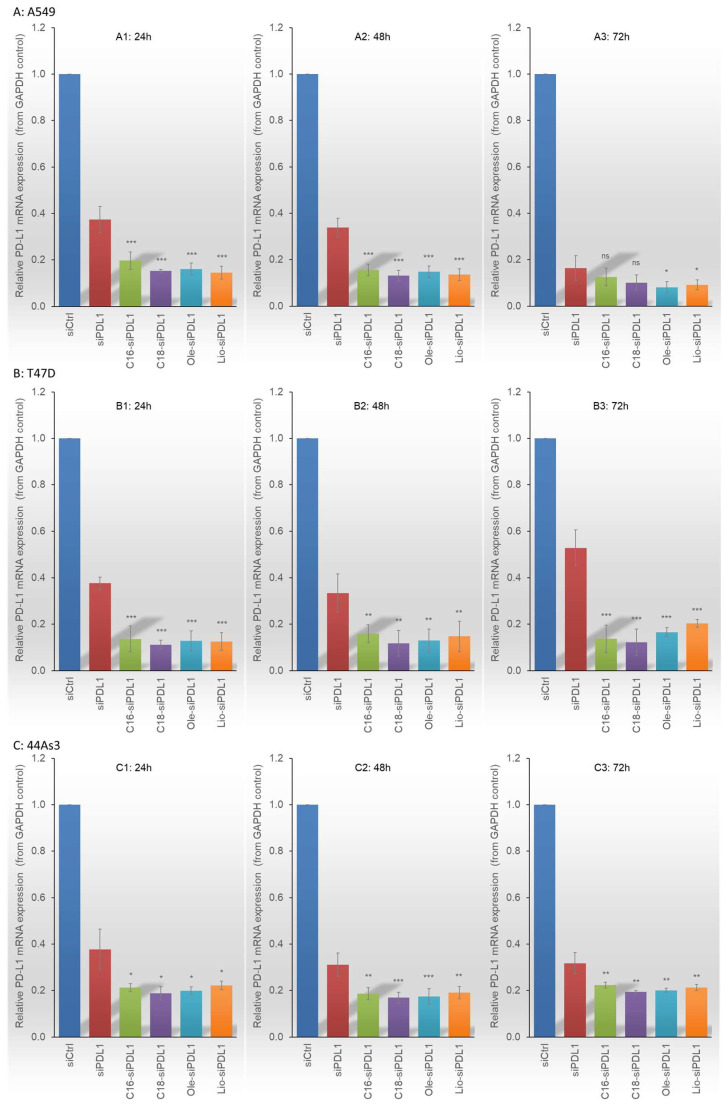
RNAi efficacies of siPDL1 and lipid-siPDL1s in A549 (**A**), T47D (**B**), and 44As3 (**C**) cell lines upon IFNγ-induced PD-L1 upregulation. Both siPDL1 and lipid-siPDL1s exhibit significant RNAi effects on cancer cells with high PD-L1 expression levels at 24–72 h post-treatment. siPDL1 and lipid-siPDL1s suppress PD-L1 expression by 60–90% in A549, 40–80% in T47D, and 60–80% in 44As3 cells compared with that in siCtrl-treated cells. Lipid-siPDL1s exert stronger RNAi effects than siPDL1 under all experimental conditions, with variations depending on IFNγ sensitivity. Data are presented as the mean ± SD of 3–5 independent experiments (ns = not significant, * *p* < 0.05, ** *p* < 0.01, *** *p* < 0.001 vs. siPDL1; *t*-test). IFNγ, interferon-gamma; PD-L1, Programmed death 1 ligand.

**Figure 6 biomolecules-15-00293-f006:**
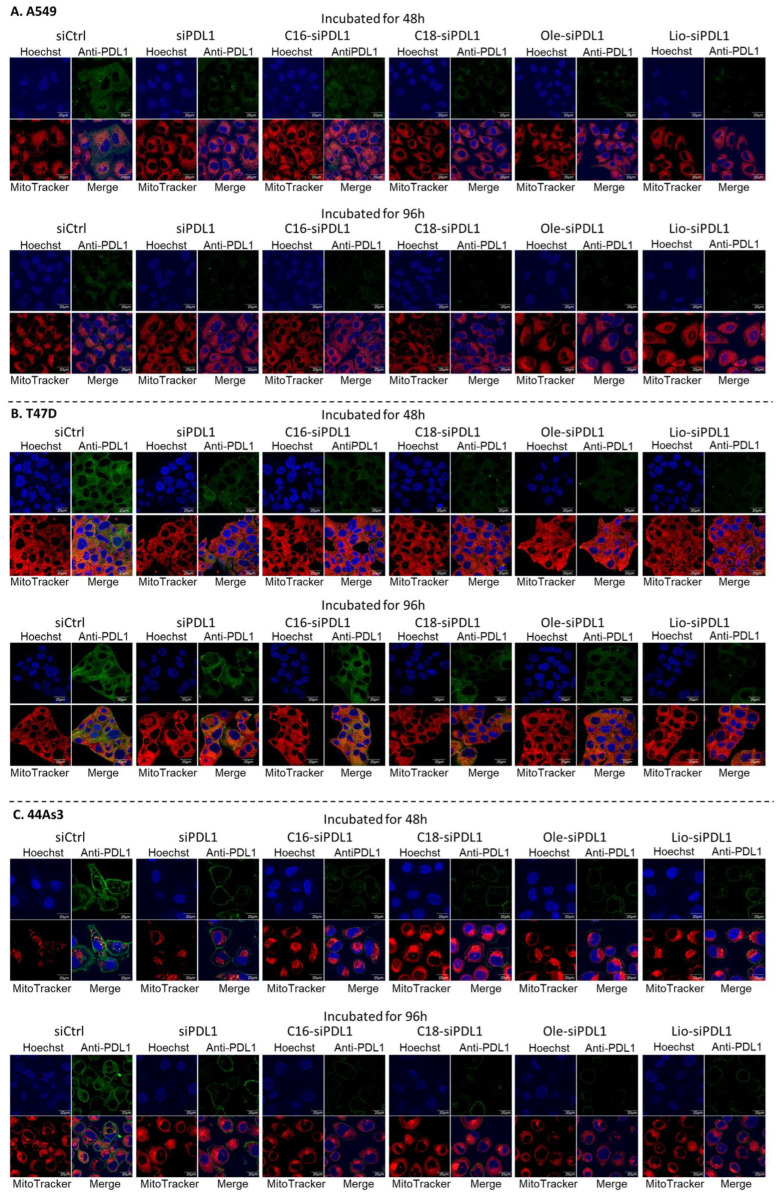
Confocal microscopy analysis of the effects of lipid-siPDL1s (50 nM) knockdown on IFNγ-stimulated PD-L1 protein expression in A549 (**A**), T47D (**B**), and 44As3 (**C**) cells after 48 and 96 h. Although cancer cells were stimulated with IFNγ (100 ng/mL) and exhibited high PD-L1 expression, siPDL1 and lipid-siPDL1s suppress the surface PD-L1 protein expression in these cells. In particular, lipid-siPDL1s strongly suppress cell surface PD-L1 protein, and the effect is long-lasting. Blue fluorescence (Hoechst33342) indicates nuclei; green fluorescence (Alexa-488 labeled anti-PD-L1 antibody) indicates PD-L1 on cancer cells; red fluorescence (MitoTracker Red) indicates mitochondria, and the merged images show the overlay of blue, green, and red fluorescence. IFNγ, interferon-gamma; PD-L1, Programmed death 1 ligand.

**Figure 7 biomolecules-15-00293-f007:**
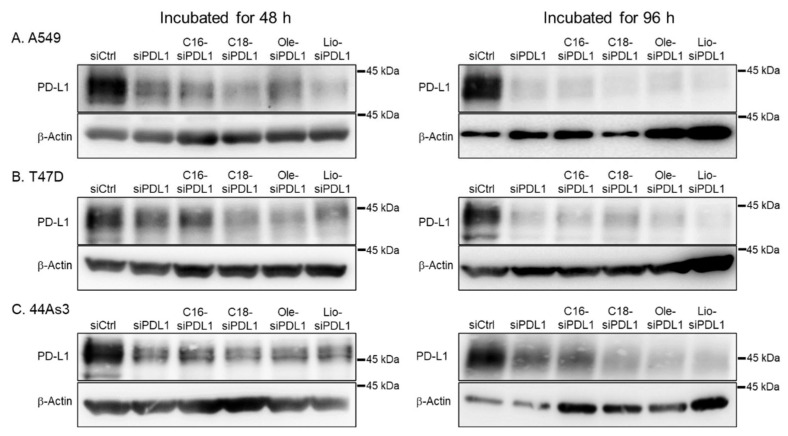
Western blot analysis of the inhibitory effect of lipid-siPDL1s on PD-L1 protein expression in A549 (**A**), T47D (**B**), and 44As3 (**C**) cells overexpressing PD-L1. Cells treated with siCtrl show strong expression of PD-L1 protein, accompanied by the detection of a strong signal; however, cells treated with lipid-siPDLs exhibit weaker expression of PD-L1 protein, and the effects of mRNA knockdown correlating with the suppression of PD-L1 protein expression. Original western blot images can be found at Appendix A.

**Table 1 biomolecules-15-00293-t001:** Characterization of ssPDL1 and lipid-ssPDL1s.

Conjugate ssRNA	Conjugated Fatty Acid	HPLC Retention Time ^a^ (min)	MALDI-TOF MS ^b^ Found/Calcd	Yield ^c^ (%)
ssPDL1	----	12.9	6704.8/6704.1	---- ^d^
C16-ssPDL1	Palmitic acid	33.9	7123.9/7121.3	80.5
C18-ssPDL1	Stearic acid	37.6	7150.4/7149.8	49.2
Ole-ssPDL1	Oleic acid	34.8	7147.2/7147.8	83.5
Lio-ssPDL1	Linoleic acid	32.8	7146.7/7145.8	62.5

^a^ Linear gradient of CH_3_CN at concentrations varying from 7 to 70% over 40 min in 100 mM TEAA (pH 7.0) using an octadecylsilyl column. ^b^ A saturated solution of 2,4,6-trihydroxyacetophenone in 50 mg/mL diammonium hydrogen citrate in 50% acetonitrile was used as the matrix. ^c^ Overall product yields were determined by measuring the absorbance at 260 nm after high-performance liquid chromatography (HPLC) purification. ^d^ Purified ssRNA was purchased. MALDI-TOF MS, matrix-assisted laser desorption/ionization-time of flight mass spectrometry.

## Data Availability

The datasets generated and/or analyzed in the current study are available in the manuscript and Appendix A.

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
