# Peer review of "Lipid-siRNA Conjugates Targeting High PD-L1 Expression as Potential Novel Immune Checkpoint Inhibitors"

_biomolecules, 2025, doi:10.3390/biom15020293_

Round 1
Reviewer 1 Report
Comments and Suggestions for Authors
This article explores the development of a novel immune checkpoint inhibitor, lipid-siPDL1s, which effectively suppresses PD-L1 expression in cancer cells. The study demonstrates that lipid-siPDL1s outperform unmodified siPDL1 and remain effective even under interferon-gamma-induced PD-L1 overexpression, suggesting their potential as an alternative cancer immunotherapy. 30% of references are relatively new in the article, 15.6% of references are self-citations.
However, I have some comments:
1) In the introduction section the author describes benefits of CPPs, such as TAT and penetratin, but there is no study of CPP in the article, please, add CPP-siRNA data into the articles or delete all mentions of CPP-siRNA conjugates.
2) "Chol-siRNA conjugates reportedly show an affinity for blood albumin, along with an improved tolerance to degradation in the blood [41]." Wolfrum et al. showed that cholesterol conjugated siRNA interacts mainly with LDL but only partially with albumin, the sentence is incorrect. Similar data about interaction of cholesterol conjugated siRNA with lipoproteins in blood were obtained by Osborn M. ( doi: 10.1093/nar/gky1232). Please, change.
3) In the last paragraph of the introduction section authors described their results in brief manner. However, the last paragraph of the introduction section should contain the main task of the article. Please, change.
4) In paragraph 3.5 there is no reference on the figures.
5) In figure 3 and figure 5 there is no SD bars in siCtrl group, why?
6) Was siRNA chemically modified (2'F, 2'O-methyl)? What kind of chemical modification were used in siRNA? The authors must show the pattern of chemical modification in the article.
7) Authors conjugated siRNA with different lipid molecules, however there is no information about using this conjugated siRNAs without transfection agent. Did author study silencing activity of lipid conjugated siRNA without transfection agent?
Author Response
Response to Reviewer 1
We thank the reviewer for their insightful comments and suggestions. We have incorporated all the recommended changes, which we believe have substantially improved the manuscript. Specific responses to the reviewers’ comments are provided below.
Reviewer 1
Comments and Suggestions for Authors
This article explores the development of a novel immune checkpoint inhibitor, lipid-siPDL1s, which effectively suppresses PD-L1 expression in cancer cells. The study demonstrates that lipid-siPDL1s outperform unmodified siPDL1 and remain effective even under interferon-gamma-induced PD-L1 overexpression, suggesting their potential as an alternative cancer immunotherapy. 30% of references are relatively new in the article, 15.6% of references are self-citations.
However, I have some comments:
1) In the introduction section the author describes benefits of CPPs, such as TAT and penetratin, but there is no study of CPP in the article, please, add CPP-siRNA data into the articles or delete all mentions of CPP-siRNA conjugates.
Response:
We thank the reviewer for this suggestion. CPP-siRNA conjugates are promising siRNA conjugates owing to their excellent cellular transfection and strong RNAi effect. We have also published a paper on siRNA with nuclear export signal peptide (NES-siRNA conjugates) (Kubo et al., Molecules, 2012, 17, 11089-11102). NES-siRNA conjugates showed stronger RNAi effects and cytoplasmic localization than non-conjugate siRNAs.
However, as the reviewer correctly pointed out, the manuscript does not present the results from a study involving peptide-siRNA conjugates. Additionally, elaborating on CPP-siRNA in the Introduction section may complicate the organization of this paper. Therefore, we have removed the description of CPP-siRNA from the text (page 2, lines 85–86, and lines 92–96 in the previous version of the manuscript). Furthermore, we have also removed relevant references (#40 and 43) from the previous version of the manuscript.
2) "Chol-siRNA conjugates reportedly show an affinity for blood albumin, along with an improved tolerance to degradation in the blood [41]." Wolfrum et al. showed that cholesterol conjugated siRNA interacts mainly with LDL but only partially with albumin, the sentence is incorrect. Similar data about interaction of cholesterol conjugated siRNA with lipoproteins in blood were obtained by Osborn M. ( doi: 10.1093/nar/gky1232). Please, change.
Response:
We thank the reviewer for these valuable remarks regarding Chol-siRNA.
We have made the following corrections on page 2, lines 87–88.
"Chol-siRNA conjugates reportedly show an affinity for low-density lipoprotein, along with an improved tolerance to degradation in the blood [40, 41]."
Additionally, the study by Osborn M., et al. (Nucleic Acids Research, 2019, 47, 1070–1081) has been added to References as #41.
3) In the last paragraph of the introduction section authors described their results in brief manner. However, the last paragraph of the introduction section should contain the main task of the article. Please, change.
Response:
We appreciate your comments regarding improvements to this manuscript.
The end of the Introduction section (page 3, lines 102–109) has been revised to address the main issue of this study as follows:
“Various cancer cell types exhibit differential expression of cancer immune checkpoints, including PD-L1, and IFNγ stimulation substantially modulates PD-L1 expression. In this study, we aimed to suppress the expression of immune checkpoint molecules, especially PD-L1, on the surface of cancer cells using RNAi technology with lipid-siRNA. Our approach was designed to be effective regardless of mRNA and protein levels, cancer cell type, or changes in expression levels upon IFNγ stimulation. This study could elucidate the potential of lipid-siRNAs as a novel immune checkpoint inhibitor based on siRNA therapeutics.”
4) In paragraph 3.5, there is no reference on the figures.
Response:
The gene knockdown % for siPDL1 and lipid-siPDL1 against each cell (A549, T47D, 44As3) compared to siCtrl-treated cells is shown in the legend of Figure 5 as follows.
“siPDL1 and lipid-siPDL1s suppress PD-L1 expression by 60–90% in A549, 40–80% in T47D, and 60–80% in 44As3 cells compared with that in siCtrl-treated cells.”
If the above statement does not meet the reviewer's comments, please highlight the relevant issue so that we can provide an appropriate response.
5) In figure 3 and figure 5 there is no SD bars in siCtrl group, why?
Response:
In this experiment, the RNAi effects of siPDL1 and lipid-siPDL1 were measured using RT-qPCR by detecting PD-L1 expression in cells treated with siCtrl as a control (the value at this time was set to “1”). Measurements are counted as n = 1 for one experiment performed in triplicate, with the experiment performed several times (n = 3‒5). All data employ siCtrl as the control (number is “1”); hence, no SD is attached to the siCtrl group.
6) Was siRNA chemically modified (2'F, 2'O-methyl)? What kind of chemical modification were used in siRNA? The authors must show the pattern of chemical modification in the article.
Response:
The siRNAs used in this experiment do not contain chemically modified nucleic acids and are all-natural siRNAs. The sequences are also presented in Figure 1 and Figure S1. In our previous studies, we have shown that lipid-siRNAs possess some resistance to nuclease compared with siRNAs (Kubo, et al., Mol. Pharmaceutics 2011, 8, 2193−2203). We have also reported that lipid-siRNA has a longer RNAi effect than siRNA (Kubo, et al., Mol. Pharmaceutics 2024, 21, 5115−5125). Furthermore, we have demonstrated that lipid-siRNAs possess superior cellular transduction properties to siRNAs (Kubo, et al., ACS Chem. Biol. 2021, 16, 150−164). Taken together, we believe that lipid-siRNAs could strongly knock down the expression of PD-L1, which is the target of this study, even with natural siRNAs.
In addition, we are currently studying lipid-siRNA with chemically modified nucleic acids and have obtained some interesting results. These results will be published in a paper once sufficient data are available. This has been addressed in the following brief comment on page 15, lines 563‒566 of the discussion section of the revised manuscript.
“Furthermore, our research group has successfully synthesized lipid-siRNA conjugates incorporating chemically modified nucleic acids, such as 2′-O-M and 2′-F. Further development of this research is expected in the future.”
7) Authors conjugated siRNA with different lipid molecules, however there is no information about using this conjugated siRNAs without transfection agent. Did author study silencing activity of lipid conjugated siRNA without transfection agent?
Response:
In all RNAi experiments, lipid-siRNAs were transfected into cancer cells using transfection agents (LFRNAi) in the current study. However, we have previously examined the RNAi effect of lipid-siRNAs in the absence of transfection agents (Kubo, et al., ACS Chem. Biol. 2021, 16, 150−164, Mol. Pharmaceutics 2012, 9, 1374−1383, Mol. Pharmaceutics 2011, 8, 2193−2203). We found that unmodified siRNAs do not exert RNAi effects in the absence of transfection agents, although high concentrations of certain lipid-siRNAs can exert RNAi effects, albeit weakly. In addition, recent studies have shown that lipid-siRNAs with longer carbon chains (e.g., behenic acid and docosahexaenoic acid) tend to show relatively stronger RNAi effects in the absence of transfection agents (Kubo, et al., ACS Chem. Biol. 2021, 16, 150−164).
In this study, we selected C16-, C18-, Ole-, and Lio-siRNAs that exhibit relatively strong RNAi effects when combined with transfection agents based on previous studies. This is briefly described in the discussion section on page 15, lines 538‒547 of the revised manuscript as follows.
“Our previous studies have revealed that lipid-siRNAs exhibit a stronger knockdown effect than unmodified siRNAs [44]. In particular, among the 16 types of lipid-siRNAs developed to date, C16-siRNA, C18-siRNA, Ole-siRNA, and Lio-siRNA exhibit potent RNAi effects [43]. Previous studies have shown that lipid-siRNAs conjugated with fatty acids containing 16−18 carbons exhibit moderate interactions with LFRNAiMAX and superior cellular transduction, indicating efficient release into cells, in contrast to lipid-siRNA conjugated with long-chain fatty acids and trans-fatty acids. Therefore, in this study, we investigated the inhibitory effects of C16-siRNA, C18-siRNA, Ole-siRNA, and Lio-siRNA on PD-L1 expression in each cancer cell line.”

Reviewer 2 Report
Comments and Suggestions for Authors
Dear Editor,
In this manuscript titled ‘Lipid-siRNA Conjugates Targeting High PD-L1 Expression as Potential Novel Immune Checkpoint Inhibitors’ authors Rina Tansou, et. al. reported the development of novel immune checkpoint inhibitor that targets PD-L1 and B7H4 using lipid-siRNA conjugates (lipid-siPDL1s). The inhibitory effect of lipid-siPDL1s on the expression of immune checkpoints (PD-L1 and B7H4) in lung (A549), breast (T47D), and scirrhous gastric (44As3) cancer cells was evaluated and the authors reported that lipid-siPDL1s exerted significantly stronger effect than unmodified siPDL1. Authors also reported that lipid-siPDL1s can strongly inhibited PD-L1 expression despite cancer cell stimulation by interferon-gamma, which induced the overexpression of PD-L1 genes. Overall, the findings about these three cancer cell lines are interesting (i.e. expression levels of immune checkpoint inhibitor proteins) and the results suggests lipid-siPDL1s could suppress PD-L1 expression for an extended duration irrespective of cancer type or PD-L1 expression level. The results are encouraging and this gene silencing technology using lipid-siRNA conjugates can be effective against all B7 family members, including PD-L1, and may serve as a new immune checkpoint inhibitor to replace conventional antibody-based inhibitors.
I recommend this manuscript to consider for publication in Biomolecules.
Thank you.
However, I just have few comments regarding the results the authors mentioned in this manuscript.
1. In their earlier publications (Ref 44, ACS chemical Biology 2021, 16, 150) authors had described the screening of 16 different fatty acid conjugates, their chemical properties, benefits of using fatty acid conjugates on sense strand and their efficacy on gene silencing. In this report most of the mRNA expressions /protein knock down data were presented up to maximum 96 hours after transfection. How long are the silencing effects active i.e. at what point the PD-L1or B7-H4 levels are back to normal compared to control? Did authors have evaluated any modifications on the anti-sense strand of PD-L1 or B7-H4 gene along with this fatty acid conjugation on the sense strand which can deliver more robust silencing (Loading more anti-sense strand into RISC complex) for a long duration beyond 72/96-hour time points?
2. In section 3.4 (IFNγ Stimulation Induces Increased PD-L1 Expression in Cancer Cells): As mentioned, figure 4A describes the increase in PD-L1 expression upon IFNγ stimulation and it shows A549 and T47D has nearly 20-folds increase compared to 44As3 cell line (which is approximate 5 folds). Whereas in Figure 4B it is mentioned that among the three cell types, 44As3 cells exhibit the highest PD-L1 mRNA expression even upon IFNγ stimulation! The graphs and the text are little confusing. So, it would be better if you please explain the difference of Figure 4A and Figure 4B more clearly.
Author Response
Response to Reviewer 2
We thank the reviewer for their insightful comments and suggestions. We have incorporated all the recommended changes, which we believe have substantially improved the manuscript. Specific responses to the reviewers’ comments are provided below.
Reviewer 2
Comments and Suggestions for Authors
Dear Editor,
In this manuscript titled ‘Lipid-siRNA Conjugates Targeting High PD-L1 Expression as Potential Novel Immune Checkpoint Inhibitors’ authors Rina Tansou, et. al. reported the development of novel immune checkpoint inhibitor that targets PD-L1 and B7H4 using lipid-siRNA conjugates (lipid-siPDL1s). The inhibitory effect of lipid-siPDL1s on the expression of immune checkpoints (PD-L1 and B7H4) in lung (A549), breast (T47D), and scirrhous gastric (44As3) cancer cells was evaluated and the authors reported that lipid-siPDL1s exerted significantly stronger effect than unmodified siPDL1. Authors also reported that lipid-siPDL1s can strongly inhibited PD-L1 expression despite cancer cell stimulation by interferon-gamma, which induced the overexpression of PD-L1 genes. Overall, the findings about these three cancer cell lines are interesting (i.e. expression levels of immune checkpoint inhibitor proteins) and the results suggests lipid-siPDL1s could suppress PD-L1 expression for an extended duration irrespective of cancer type or PD-L1 expression level. The results are encouraging and this gene silencing technology using lipid-siRNA conjugates can be effective against all B7 family members, including PD-L1, and may serve as a new immune checkpoint inhibitor to replace conventional antibody-based inhibitors.
I recommend this manuscript to consider for publication in Biomolecules.
Thank you.
However, I just have few comments regarding the results the authors mentioned in this manuscript.
- In their earlier publications (Ref 44, ACS chemical Biology 2021, 16, 150) authors had described the screening of 16 different fatty acid conjugates, their chemical properties, benefits of using fatty acid conjugates on sense strand and their efficacy on gene silencing. In this report most of the mRNA expressions /protein knock down data were presented up to maximum 96 hours after transfection. How long are the silencing effects active i.e. at what point the PD-L1or B7-H4 levels are back to normal compared to control? Did authors have evaluated any modifications on the anti-sense strand of PD-L1 or B7-H4 gene along with this fatty acid conjugation on the sense strand which can deliver more robust silencing (Loading more anti-sense strand into RISC complex) for a long duration beyond 72/96-hour time points?
Response:
We appreciate your valuable comments. We also thank you for your careful review of our previous paper.
Lipid-siRNA conjugates can maintain RNAi effects for a substantially prolonged period compared with unmodified siRNAs. For example, in our recent paper (Kubo, et. al., Mol. Pharmaceutics 2024, 21, 5115–5125), we reported that unmodified siRNA exerted almost no RNAi effect after approximately 10 days of cell transfection, whereas lipid-siRNA knocked down approximately 50% of target genes even after 13 days of cell transfection. In another study, we confirmed that lipid-siRNAs are more resistant to nuclease degradation than unmodified siRNAs (Kubo, et. al., Mol. Pharmaceutics 2011, 8, 2193−2203). In addition, lipid-siRNA exhibited superior cell transfection properties compared to unmodified siRNAs (Kubo, et. al., ACS Chemical Biology 2021, 16, 150). Based on these results, we consider that the lipid-siRNAs in this study can also maintain the RNAi effect for a longer period than unmodified siRNAs. However, the precise time period for which lipid-siRNAs exhibit a sustained RNAi effect is yet to be determined. Various factors are involved, including differences in RNAi effects due to siRNA sequences, differences in target genes, and differences in lipids to be conjugated, and these factors must be included and examined.
Additionally, as highlighted by the reviewer, it is very interesting to introduce chemically modified nucleic acids into the sequence. In fact, we have produced lipid-siRNAs with chemically modified nucleic acids and are currently investigating their RNAi effects, with some interesting results. These results will be published once sufficient data are available. This is addressed in the following brief comment on page 15, lines 563‒566 of the discussion section of the revised manuscript.
“Furthermore, our research group has recently achieved success in synthesizing lipid-siRNA conjugates incorporating chemically modified nucleic acids, such as 2′-O-M and 2′-F. Further development of this research is expected in the future.”
- In section 3.4 (IFNγ Stimulation Induces Increased PD-L1 Expression in Cancer Cells): As mentioned, figure 4A describes the increase in PD-L1 expression upon IFNγ stimulation and it shows A549 and T47D has nearly 20-folds increase compared to 44As3 cell line (which is approximate 5 folds). Whereas in Figure 4B it is mentioned that among the three cell types, 44As3 cells exhibit the highest PD-L1 mRNA expression even upon IFNγ stimulation! The graphs and the text are little confusing. So, it would be better if you please explain the difference of Figure 4A and Figure 4B more clearly.
Response:
All cancer cells used in this study (i.e., A549, T47D, and 44As3) exhibited PD-L1 expression upregulated by several-fold upon IFNg stimulation. Figure 4A shows the IFNγ concentration-dependent changes in PD-L1, compared with their baseline levels without IFNγ exposure (0 ng/mL). As shown in Figure 4B, we analyzed PD-L1 expression in each cell upon IFNγ (100 ng/mL) stimulation relative to A549 cells stimulated with IFNγ (100 ng/mL). Given that the differences between Figure 4A and Figure 4B were unclear in the previous description, this has been rewritten in the revised manuscript (page 9, lines 365‒376) as follows.
“Notably, A549 and T47D cells, which originally expressed relatively low PD-L1 levels, showed up to a 20-fold increase in PD-L1 expression following IFNγ stimulation at 100 ng/mL concentration. Furthermore, 44As3 cells, which originally expressed high PD-L1 levels, showed up to a 5-fold increase in PD-L1 expression under the same conditions (Figure 4A). This difference in sensitivity to IFNγ between cancer cells is dependent on the baseline PD-L1 expression level. Specifically, PD-L1 expression in 44As3 cells, which initially exhibit high PD-L1 levels, does not fluctuate greatly upon IFNγ stimulation, whereas A549 and T47D cells, which initially have low PD-L1 levels, show a significant increase in PD-L1 expression in response to IFNγ stimulation. Interestingly, 44As3 exhibited the highest PD-L1 expression among the three cancer cell lines following IFNγ stimulation at 100 ng/mL with levels approximately 15-fold higher than those observed in A549 and T47D cells (Figure 4B).”

Reviewer 3 Report
Comments and Suggestions for Authors
The manuscript by Tansou et al. investigates synthesized Lipid-siRNA conjugates to suppress PD-L1 expression as potential inhibitors. The overall manuscript is well-written, and most data are scientifically well presented and explained. Here are some important points that authors need to address:
- For PAGE analysis and western blot data, please indicate the size/marker in the main figures.
- The original MALDI-TOF MS data is missing.
- There are C16-, C18-, Ole-, and Lio- as lipid conjugators. The current data show they have very similar functions. The authors should explain more about this observation and the indication behind it.
- The main experiments are checking PD-L1 expression by RT-PCR and protein expression. There is no phenotype experiment, and the focus only on affecting protein expression, which makes it very hard to believe how much effect it has as inhibitors. Decreased protein expression levels cannot directly indicate the potential increased effect of being killed by T cells.
- When comparing the inhibition effects between siPDL1 and Lipid-siPDL1s, like in figure 5, it is obvious there is a significant difference, but the effect is 60% vs. 80%, sometimes even 70% vs. 80%. Having a significant difference does not mean having a big difference, especially as the authors want to show the increased effects of Lipid-siPDL1s. More experiments, analysis, and discussion are necessary to show the benefits of Lipid-siPDL1s.
Author Response
Response to Reviewer 3
We thank the reviewer for their insightful comments and suggestions. We have incorporated all the recommended changes, which we believe have substantially improved the manuscript. Specific responses to the reviewers’ comments are provided below.
Reviewer 3
Comments and Suggestions for Authors
The manuscript by Tansou et al. investigates synthesized Lipid-siRNA conjugates to suppress PD-L1 expression as potential inhibitors. The overall manuscript is well-written, and most data are scientifically well presented and explained. Here are some important points that authors need to address:
1) For PAGE analysis and western blot data, please indicate the size/marker in the main figures.
Response:
We appreciate your comments regarding improving our manuscript.
PAGE analysis (Figure 1D) was re-performed, including markers. The marker size is noted in the figure; Figure 1D is submitted as a revised figure.
Protein markers are used in all western blot analyses. These data are listed in the original western blot images in the Supporting Material. Protein markers were detected using Epi-white light, while PD-L1 and β-actin proteins were detected using chemiluminescence. Therefore, original western blot images were created by overlaying these two images in the Supporting Material. In the revised Figure 4C and Figure 7, protein size is shown in the figure based on the overlaying image in Supporting Material.
2) The original MALDI-TOF MS data is missing.
Response:
The original MALDI-TOF MS data for ssPDL1, lipid-ssPDL1s, ssB7H4, and lipid-ssB7H4s are presented in revised Figure S2 in Supporting Material.
3) There are C16-, C18-, Ole-, and Lio- as lipid conjugators. The current data show they have very similar functions. The authors should explain more about this observation and the indication behind it.
Response:
For the C16-, C18-, Ole-, and Lio-siRNA conjugates used in this study, lipid-siRNAs that have shown relatively strong RNAi effects in previous studies were selected and used to suppress PD-L1 expression. In our previous study, we synthesized 16 different types of lipid-siRNAs and conducted a detailed investigation of their RNAi effects (Kubo, et al., ACS Chemical Biology 2021, 16, 150). We found that conjugation of siRNA with 16‒18 carbon fatty acids enhanced the RNAi effect. We proposed that lipid-siRNA conjugated with 16‒18 carbon fatty acids has a moderate affinity for lipofectamine, a cell transduction agent, and cell membranes and is also moderately released in cells. Similarly, in this study, C16-, C18-, Ole-, and Lio-siRNAs interacted moderately with lipofectamine and cell membranes and were released smoothly in the cell compared with long-chain fatty acids and trans-fatty acids.
A brief description has been added to the discussion section of the revised manuscript (page 15, lines 538‒547) as follows.
“Our previous studies have revealed that lipid-siRNAs exhibit a stronger knockdown effect than unmodified siRNAs [44]. In particular, among the 16 types of lipid-siRNAs developed to date, C16-siRNA, C18-siRNA, Ole-siRNA, and Lio-siRNA exhibit potent RNAi effects [43]. Previous studies have shown that lipid-siRNAs conjugated with fatty acids containing 16−18 carbons exhibit moderate interactions with LFRNAiMAX and superior cellular transduction, indicating efficient release into cells, in contrast to lipid-siRNA conjugated with long-chain fatty acids and trans-fatty acids. Therefore, in this study, we investigated the inhibitory effects of C16-siRNA, C18-siRNA, Ole-siRNA, and Lio-siRNA on PD-L1 expression in each cancer cell line.”
4) The main experiments are checking PD-L1 expression by RT-PCR and protein expression. There is no phenotype experiment, and the focus only on affecting protein expression, which makes it very hard to believe how much effect it has as inhibitors. Decreased protein expression levels cannot directly indicate the potential increased effect of being killed by T cells.
Response:
As highlighted by the reviewer, this study focused on demonstrating that lipid-siRNA could suppress PD-L1 expression in cancer cells at the mRNA and protein levels. Further experimental verification is needed to determine whether the observed reduction in PD-L1 mRNA and protein expression improves T cell killing ability. However, Ganesh et al. recently reported the in vivo anti-tumor effect of siPDL1-lipid conjugate (the structure of this conjugate differs from that of the conjugate examined in our study), indicating its potential as an immune checkpoint inhibitor (Molecular Therapy, 32, 2024).
Although the results of this study are basic, our lipid-siRNAs have sufficient potential as immune checkpoint inhibitors and will be further investigated, including in vivo anti-tumor effects.
The following statement has been added to the Discussion section on page 15, lines 558‒563, describing the work of Ganesh et al. on this topic (Ref #51).
“Recently, Ganesh et al. investigated an in vivo combination therapy using standard-of-care immune checkpoint inhibitors alongside chemically stabilized acylated siRNAs targeting signal transducer and activator of transcription 3 and PD-L1 genes, reporting excellent antitumor effects [51]. This study provides substantial evidence supporting the potential of siRNA conjugates, including lipid-siRNAs, as novel immune checkpoint inhibitors targeting immune checkpoints such as PD-L1."
5) When comparing the inhibition effects between siPDL1 and Lipid-siPDL1s, like in figure 5, it is obvious there is a significant difference, but the effect is 60% vs. 80%, sometimes even 70% vs. 80%. Having a significant difference does not mean having a big difference, especially as the authors want to show the increased effects of Lipid-siPDL1s. More experiments, analysis, and discussion are necessary to show the benefits of Lipid-siPDL1s.
Response:
As highlighted by the reviewer, further experimental approaches are needed to demonstrate the effects of lipid-siRNAs. We have successfully developed a new lipid-siRNA with chemically modified nucleic acids such as 2′-O-Me and 2′-F in the sequence to further enhance this lipid-siRNA. Currently, we are collecting data on multiple aspects of this new lipid-siRNA incorporating chemically modified nucleic acids and hope to report our findings in a scientific paper in the near future.
This is addressed in the following brief comment on page 15, lines 563‒567 of the discussion section of the revised manuscript.
“Furthermore, our research group has recently achieved success in synthesizing lipid-siRNA conjugates incorporating chemically modified nucleic acids, such as 2′-O-M and 2′-F. Further development of this research is expected in the future.”

Round 2
Reviewer 3 Report
Comments and Suggestions for Authors
The authors have effectively addressed all of my comments.